# Nutrition Transition in the Post-Economic Crisis of Greece: Assessing the Nutritional Gap of Food-Insecure Individuals. A Cross-Sectional Study

**DOI:** 10.3390/nu11122914

**Published:** 2019-12-02

**Authors:** Eleni Chatzivagia, Aleks Pepa, Antonis Vlassopoulos, Olga Malisova, Konstantina Filippou, Maria Kapsokefalou

**Affiliations:** Department of Food Science & Human Nutrition, Agricultural University of Athens, Iera Odos 75, 11855 Athens, Greece; chatzivagiaeleni@gmail.com (E.C.); alekspepa@gmail.com (A.P.); avlassopoulos@aua.gr (A.V.); olgamalisova@yahoo.gr (O.M.); filippoukonstandina@gmail.com (K.F.)

**Keywords:** food security, malnutrition, food provision, nutritional status, FEAD

## Abstract

Food insecurity has risen by 40% in Europe’s post-economic crisis, linked to the economic turmoil and austerity. Despite the intensification of efforts to fight all forms of poverty, including the implementation of programs targeted to the most deprived, the study of individuals at risk of food insecurity has been largely neglected. This study aimed to map the nutritional habits and needs of the most deprived in Greece, one of the countries most affected by the economic crisis. Individuals classified as most deprived under the Fund for the European Aid to the Most Deprived (FEAD) criteria (*n* = 499) from across Greece and an age matched control from the general population (*n =* 500) were interviewed between December 2017 and December 2019. Participants provided information about demographic characteristics, and self-reported anthropometric measures and nutritional intake of the past month via a food frequency questionnaire (FFQ). Protein and energy malnutrition were defined as daily intake <1.950 kcal and ≤0.75 g/kg body-weight accordingly. Protein and energy malnutrition were high among FEAD recipients (52.3% and 18.6% respectively, *p* < 0.001), alongside a high prevalence of overweight and obesity (BMI > 25: 68.4% versus 55.1%; *p* < 0.001). The diet of FEAD recipients included higher amounts of carbohydrates, lower amounts of monounsaturated fat (MUFA) and polyunsaturated fat (PUFA; *p* < 0.001 compared to control), larger amounts of plant-based proteins (5.81 ± 1.7 versus 4.94 ± 1.3% E respectively, *p* < 0.001) and very limited intake of fish (0.07 portions/day). Despite being enrolled in a food assistance program, protein and energy malnutrition is prevalent among Greece’s most deprived who experience not only lower diet quality but also the double burden of malnutrition.

## 1. Introduction

‘Food security exists when all people, at all times, have physical, social and economic access to sufficient, safe and nutritious food that meets their dietary needs and food preferences for an active and healthy life’ [1], is a definition from the Food and Agriculture Organization (FAO) in 1974. More than 40 years later, the elimination of food insecurity and all forms of malnutrition worldwide are still a focal point of the Sustainable Development Goals [2]. Food insecurity describes a spectrum of conditions characterized by reduced access to food, from worrying and uncertainty about the ability to obtain food (mild food insecurity); to a reduction in the quality, quantity and/or frequency of having access to food (moderate food insecurity); and to the establishment of hunger and the inability to secure food for a day or longer (severe food insecurity) [3].

Although severe food insecurity has a prevalence of approximately 1.5% in Western countries, mild to moderate food insecurity affects 10%–15% of the population [4]. In response to the problem, public health policies have been developed in order to help the food-insecure population stretch their food budgets and buy healthy food [5]. For example, the United States of America dedicated public health programs to all stages of life specifically aimed to reduce food insecurity, such as the Supplemental Nutrition Assistance Program (SNAP) [6]. In Europe, food security has not been always part of the public health agenda. After the beginning of austerity (circa 2009) though, with 118.0 million people in the EU (23.5% of the population) living in households at risk of poverty or social exclusion [7], the decreasing trend of food insecurity was reversed. Ever since poverty reduction is one of the headline goals of the Europe 2020 strategy [8]. The Fund for European Aid to the Most Deprived (FEAD) has been set up to contribute to achieve this goal. Its specific objective is to alleviate the worst forms of poverty in the EU and to promote the social inclusion of the most deprived persons [9]. It provides material support and social inclusion measures to each time targeted population group across the EU, but the evaluation of its effectiveness is still to be reported. FEAD supports in European countries actions to provide food and/or basic material assistance to the most deprived; non-material assistance may also be provided in order to help them integrate better into society [9]. National authorities organize the delivery of the assistance through partner organizations selected on the basis of objective and transparent criteria defined at the national level [10].

Greece was the country most severely affected by the economic downturn and ensuing austerity. According to Eurostat, in 2012 about 3.8 million people in Greece (34.6% of the total population) were at risk of poverty or social exclusion, while unemployment reached 26% at the end of 2012 [11]. These changes in socioeconomic status were reflected to a rise in all-age mortality between 2000–2016 [12] and triggered humanitarian action to help those most in need.

Since 2015, in Greece, FEAD has been the only nationwide program providing food aid and basic material assistance to everyone living under extreme poverty (around 400,000), including those who are homeless [13]. In order to be enrolled in FEAD, one should fulfill a list of socioeconomic criteria decided by the Ministry of Labor, Social Security, and Social Solidarity on a central government level, including employment status, income, marital status, number of children, etc. [14]. In Greece, every potential recipient can submit an online application and after a cross check with the economic data, a finalized list of beneficiaries is formed, which is updated on a monthly basis.

Despite the willingness to address food insecurity through the alleviation of poverty, those efforts are hindered by the lack of comprehensive data on the magnitude of the problem and the type of support needed. The study of food insecurity and undernutrition is a relatively neglected topic in public health research in Europe [15]. Across Europe, the European Quality of Life Survey is the only source of data on food insecurity but the survey itself has only one question dedicated to the topic. [16] Although national nutrition and health surveys are a suitable source to derive food insecurity and malnutrition data (given that they are repeated frequently on nationally representative population samples), they do not cover the need for a clear description of the profile and needs of the people living under the risk of food insecurity [15]. Unfortunately, such data are scarce, with implications on the prevention and management of this growing public issue.

This study aimed to assess the nutritional intake and dietary habits of FEAD recipients in Greece with the objective to better understand the needs and habits of this vulnerable population group and help improve local policies and public health actions in the framework of the FEAD program in Greece.

## 2. Methodology

### 2.1. Research Design and Inclusion Exclusion Criteria of Study Sample

A cross sectional study was carried out during December 2017–December 2018. The Ethics Committee of the Agricultural University of Athens approved the design, the procedures and the aim of the study. A consent form was given to the participants to inform them about the content of the survey, the anonymity of the questionnaires and the safeguarding of personal data based on the GDPR standards.

A total of 499 recipients of the FEAD program across all Greece were recruited in the study (cases-FEAD recipients) alongside a control group with similar age demographics (*n =* 545). Cases questionnaires were collected at FEAD delivery points at the time of food provision distribution, and were limited to people having received at least one more food provision delivery. Subjects had to be able to communicate in Greek and provide information without requiring a proxy. Migrants were recruited given they could communicate in Greek on their own or through a proxy. Participants for the general population group, designed to mimic the geographic distribution of the FEAD group, were recruited in parks, squares, outside schools or at recreational spaces, and were mapped in order to match the age demographic of the cases. We also made sure that the general population group did not include any FEAD recipients. All the collections were conducted in both urban and rural areas. More specifically, individuals receiving food assistance under FEAD from five different areas of Greece were interviewed, including the capital and smaller cities (peripheries: Attica, West Macedonia, Central Macedonia, Crete and Peloponnese). In total, 72 municipalities were included in the study, with representation from urban and rural locations, as they represent 66% of the total Greek population. We covered mainland Greece and the largest island. Smaller islands were not included due to difficulty in access and a low participation in the FEAD program. All data were collected via a questionnaire which collected self-reported data on socio-demographic characteristics, dietary habits and anthropometry via a 20 min interview with a trained nutritionist. The recruitment process followed a one-step process in which participants were checked against the eligibility criteria and enrolled in the study at the same.

### 2.2. Socio-Demographic Characteristics

Socio-demographic variables that were recorded were: gender, age, educational level measured by years of school, number of children, number of people living in the household, occupational status (in the following categories: employed, unemployed, retired and housewife), and marital status categorized as single, married, divorced or widowed.

### 2.3. Anthropometry and Dietary Assessment

Body weight (in kilograms) and height (in meters) were recorded as self-reported values. Body mass index was then calculated as weight (in kilograms) divided by height (in meters squared). Overweight and obesity were defined as body mass index 24.9–29.9 kg/m^2^ and >29.9 kg/m^2^, respectively according to WHO criteria [4].

Dietary habits were assessed through a validated semi-quantitative [17] food frequency questionnaire (FFQ) that evaluates dietary intake over the past month. Briefly, the FFQ includes 38 questions targeted to assess the frequency of consumption of all main food groups (i.e., dairy products, cereals, fruits, vegetables, meat, fish, legumes, added fats, alcoholic beverages and sweets), in a 6-grade scale ranging from never to more than twice daily, alongside questions on dietary behaviors (i.e., eating in restaurants or canteens, consumption of breakfast, number of meals consumed on a daily basis and daily water consumption).

The data from the FFQ were analyzed according to the following process. Consumption of all foods/food groups was calculated as daily consumption in g or mg via a multiplication of the standard serving size of each food group (one per question) defined in the questionnaire by a factor corresponding the frequency of consumption (frequency factors: never = 0; 1–3 times/month = 0.07; 1–2 times/week = 0.21; 3–6 times/week = 0.64; 1 times/day = 1; 2 times/day = 2).

Intakes of energy, macro and micronutrients were then calculated using two complementary food composition databases for generic and culture specific food items [18,19]. Macronutrient intakes were calculated both as absolute values and as percentages of total energy. Total protein consumption was classified by source into protein from plant sources and protein from animal sources. Protein from plant sources included vegetable mains, vegetable salads, fruits, nuts, legumes, potatoes, bread, cereal, rice pasta, fruit juices and olive oil variables. Protein from animal sources included milk yogurt, milk yogurt light, cheese, egg, beef, pork, poultry, lamp, cold cuts, fish and other seafood and oil fat variables. Total fat was further distinguished into polyunsaturated (PUFA), monounsaturated (MUFA) and saturated fat (SFA), as percentages of energy.

The primary focus of this analysis was the detection of protein and energy malnutrition. More specifically, energy malnutrition was evaluated against an energy cutoff of a minimum daily intake <1.950 kcal, recommended by the FAO Statistics Division (minimum dietary energy requirement—Greece) [20]. Energy deficit was calculated as the distance between the current energy intake of each individual and the FAO cut off. Protein malnutrition was defined as daily intake ≤0.75 gkg body-weight [20].

Differences in the dietary habits between the two groups were evaluated using the absolute intakes of foods/food groups and nutrients. Dietary quality was assessed using percentages of the population/group achieving the recommended daily intakes. For fat, protein and carbohydrate consumption, the recommendations from FAO/WHO were used [21]. For fiber the adequate intake value by European Food Safety Authority (EFSA) (25 g per day) was used [22], while for calcium the population reference intakes from the same source were used (950 mg per day) [22]. The reference value used for sodium was the recommendations of Greek Food-Based Dietary Guide’s (GFBDG) (2.300 mg per day) [23], which refers to both sodium that exists in food and sodium added as table salt or while cooking. Intakes of the food groups were assessed against the GFBDG and the serving recommendations for the consumption of each food group. An additional comparison was conducted with the WHO East Mediterranean Region’s recommendations of servings for a healthy diet [24]. The recommendations of both guidelines are presented in the Appendix A. Briefly, from both guidelines information about the intake of 8 major food groups (vegetables, fruits, legumes, cereal, dairy, meat, fish and oils) was collected on the frequency of consumption and the recommended daily or weekly intake. For foods that were recommended for weekly intake, the equivalent portion per day (*equivalent daily portion = (recommended frequency × recommended portion per eating occasion)/7*) was calculated for the statistical analysis.

### 2.4. Statistical Analysis

Normally distributed continuous variables were presented as mean values ± their standard deviations (mean ± SD), and categorical variables as absolute and relative frequencies separately for cases and controls. P-P plots and histograms were used to assess normality. Independent sample *t*-tests and Mann–Whitney U-tests were used to determine differences between variables. Differences between groups were calculated with chi squared tests. All reported *p*-values were compared to a significance level of 5%. The IBM SPSS Statistics 23.0 statistical software package was used for analyzes.

## 3. Results

The basic anthropometric, socio-demographic and lifestyle characteristics of FEAD recipients and the general population are presented in Table 1. There was no difference in the age of the participants in the two population groups (47.53 ± 13.1 versus 47.82 ± 13.6). Overall, the majority of participants were married in both groups, but unemployment was higher among FEAD recipients (76%) compared to the general population (16%, *p* < 0.001). FEAD recipients had less years of education (10.98 ± 8.5 versus 12.66 ± 3.6, *p* < 0.001) and were more likely to have two or more children (*p* < 0.001). Being a FEAD recipient was associated with higher prevalence of overweight and obesity (overweight 44.0% versus 37.5% and obesity 25.4% versus 18.0% in FEAD versus general population) with barely a quarter of the FEAD recipients being classified within the normal BMI range (28.1%). Being female was associated with lower obesity rates among FEAD recipients (39.0% versus 51.15%, female versus male), contrary to the findings in the general population (48.1% versus 27.8%, female versus male).

The diet of FEAD recipients included higher proportions of carbohydrates and lower proportions of fats (as percentages of total energy). Especially for fats, FEAD recipients reported a diet higher in PUFA and lower in MUFA than the general population (*p* < 0.001). The SFA intake was similar for both populations and higher than the recommended 10% of total energy cut-off (Table 2). Similarly, the amount of energy coming from proteins where higher for FEAD recipients despite a lower intake of protein as g per kg body weight (Table 2). Sex specific differences were also seen for protein intake, as male FEAD recipients consumed less protein (g/kg) than females (1.2 (0.82, 1.84) versus 1.5 (1.01, 2.33)) while the opposite was seen in the general population (1.68 (1.23, 2.12) versus 1.38 (0.97, 2.09), males versus females) (*p* < 0.001 for all, data not shown).

Protein sources were also different among the two populations as FEAD recipients reported higher intakes of plant proteins (3.35% ± 1.5% of daily energy) compared to the general population (2.80% ± 1.3% of daily energy); the opposite was seen for animal protein (*p* < 0.001 for both).

FEAD recipients consumed higher amounts of fiber and calcium, and lower amounts of sodium (Table 2). However, still, only 59.9% of the FEAD recipients reached the recommended daily fiber intake (compared to 51.4% of the general population, *p* < 0.001, data not shown).

Analysis of the food groups’ intakes highlighted differences in most food groups between FEAD recipients and the general population. FEAD recipients consumed smaller amounts of fruit juices, oils and nuts, meat, fish and seafood, confectionary and ice cream and alcohol than the general population. At the same time, consumption of legumes and potatoes was higher among the FEAD recipients (Table 3, Figure 1).

FEAD recipients were more likely to have 1–3 meals per day compared to 4–5 meals per day for the general population (Figure 2A, *p* < 0.001). FEAD recipients reported skipping breakfast more often than the general population (58.8% versus 38.8%) (Figure 2B).

FEAD recipients consumed less total energy, and lower amounts of protein per kg body weight (*p* < 0.001) (Table 2). Applying the 1950 kcal per day cut off, only 58% of FEAD recipients reported consuming adequate energy compared to 77% of the general population (Figure 3A). The energy deficit (distance to achieve the 1950 kcal/day cut-off) for each group is illustrated in Figure 3B. Between genders, analysis for energy malnutrition revealed no difference. FEAD recipients were more likely to experience protein malnutrition defined as protein intake below ≤ 0.75 g/kg body-weight (18.6% versus 5.0% in the general population, Figure 4, *p* < 0.001). Insufficient protein intake was more common among males in the FEAD recipients and vice versa for the general population (Figure 4, *p* < 0.001).

## 4. Discussion

This study is one of the first attempts to assess the eating habits of the most deprived after the economic crisis in Greece. Our results show that almost ten years after the start of the economic crisis, a significant and alarming proportion of those enrolled in a food assistance program are still experiencing protein and energy malnutrition. These people not only have lower energy and protein intakes but are also more likely to skip meals, especially breakfast, and report lower consumptions of meat, fish and seafood, oils and nuts.

Our analysis identified 42% of the FEAD recipients and 23% of the general population as having inadequate energy intake at the time of the survey. Interestingly, at the same time, the prevalence of underweight was measured just over 2% in the same population while an astonishing 69.4% of the FEAD recipients were classified as overweight/obese, characteristic of the double burden of malnutrition. These findings are in agreement with previous studies. For example, in the US, similar patterns were observed among recipients of food assistance programs [25], especially women [26,27], and there was and a clustering of increased BMI with lower household income and educational level, both traits common among FEAD recipients [28].

In terms of dietary choices, FEAD recipients exhibited a shift from animal to plant based products with the associated changes in carbohydrate and fiber intake and a shift in their protein sources. Unfortunately, the diets of FEAD recipients also included lower intakes of MUFA, through the reduction of olive oil use, despite it being a Mediterranean diet staple. Despite higher intakes of total PUFA among FEAD recipients it is important to highlight the concurrent, very small intake of fish which barely reaches 0.5 portions per week for the average FEAD recipient (three times lower than the general population’s average fish and seafood intake), indicating a potentially higher intake of vegetable oils. At this point, it is important to mention that only 9.4% of FEAD recipients reached the recommended weekly intake of 1–2 portions of fish compared to 15.4% of the general population (data not shown). However, poor dietary habits were also seen for fruits and vegetables with barely 4.2% of the FEAD recipients reaching the amount recommended by WHO’s Eastern Mediterranean office of five portions of vegetables per day, and no-one reaching the recommended four servings of fruit. Equally, poor adherence to the guidelines were seen for the general population (0% for fruit and 16.1% for vegetables). However, in FEAD recipients, we observed higher fiber intake mainly through the increased intake of legumes (three fold higher adherence to the recommended legumes intake compared to control) and lower intake of red meat without any impact on the intake of animal protein (see Appendix A for the adherence to guidelines analysis). The sustained high intake of meat products, confectionary and ice cream could also be a possible explanation for the high intake of SFA, which exceeded the 10% of energy cut off for both groups. Previous research has shown that such dietary habits are common among low socioeconomic status individuals and those receiving food assistance and could be linked with their purchasing power and the added cost of a ‘healthy’ diet [4,29,30,31,32,33,34,35,36,37]. More specifically for the Mediterranean region, previous research [38,39] has indicated a link between the economic downturn and the decline of Mediterranean diet adherence within the general population, leading to inequities in food access.

Regarding salt intake, only 2% of FEAD recipients and 4% of general population exceeded the guidelines. Food sodium intake was low in both the food insecure group and control from the general population (1037.9 ± 509.3 versus 1245.29 ± 531.81 mg/day) compared to another study [29] of the Greek population reporting higher intakes (1983.2 ± 814.1 mg/day). This discrepancy could be linked to the fact that salt intake in this study did not include table salt or salt added while cooking, accounting for 10%–15% of salt intake [30,31,32]. At the same time the salt content of foods could also be underestimated, linked to the lack of updates of the Greek Food Composition Table [33]. The need for a unified approach and an updated FFQ for salt intake in Greece should be highlighted for future research.

When asked about breakfast consumption, the clear majority (71.4%) reported skipping daily breakfast. This is in agreement with findings from the Food Foundation and the End Hunger UK Coalition, showing that 16% of adults and 23% of parents in the UK are skipping meals out of financial necessity [34]. These indicate that the food assistance programs in Greece should start considering the true nutritional needs of this vulnerable population, and maybe what is lacking is not food aid in general but a specific target in meals and food groups.

Overall, this study sheds light on the needs and habits of Greece’s most vulnerable, showing that the economic crisis could have initiated a “nutritional transition” in the country affecting mainly those less privileged. Such findings as the nutritional transition characterized by lower adherence to the Mediterranean diet and the existence of the double burden of malnutrition shown by our results are in agreement with reports of the FAO for the Mediterranean region [40]. Under these circumstances more research is needed to better understand the needs and habits of those vulnerable groups in order to design public health interventions that will not only provide access to food but will make the healthiest choice the easiest choice, in order to shape healthy food patterns, especially at a time of economic crisis [41,42].

This study is not free of limitations. Firstly, we need to mention the fact that FEAD recipients were selected to be individuals that had received at least one food provision delivery. This choice was made to ensure that study participants were at risk of food insecurity for more than a month (the national FEAD register is updated monthly) in order to exclude individuals with very short-term, transient financial difficulties that may opt for the FEAD program. This choice resulted in a study sample with a more homogeneous risk for food insecurity due to financial hardship but at the same time made it impossible for the researchers to measure the ‘true’ dietary needs of those living under the risk of food insecurity in the absence of any form of financial or material support. Along those lines it is also important to mention that FEAD participation is mutually exclusive to other state ran food security initiatives; FEAD recipients could have access to other forms of food assistance (church, soup kitchens, NGOs, food banks, etc.). However, although this makes the use of these data for the evaluation of the FEAD program problematic, it does not present as an issue for their intended use, which is the evaluation of the nutritional status of those most deprived. Although the current study covered a representative sample of the Greek population with recruitment taking place both in rural and urban locations, the use of questionnaires only available in Greek could have excluded beneficiaries not fluent in the language. However, according to the FEAD Midterm Report in Greece, only 1% of the beneficiaries are migrants [9]. It is worth mentioning at this point that FEAD is not targeted towards asylum seekers and recent immigrants but only those who have formally completed all processes and have a national insurance number.

Moving on to the more technical limitations, the heights and weights of all participants were self-reported with the associated biases and errors [43,44]; however, when compared to the national nutrition and health survey which used objective measures, we did not identify large differences between the two [45]. The second source of error is linked with the choice of dietary assessment methods. The choice of an FFQ, although suitable for the purpose of the study, does not allow for a detailed analysis of the specific foods consumed, their source, etc. A combination of FFQ and 24 h recalls, ideally from multiple days, could provide more detailed results. As a form of qualitative research, focus groups can be used, in order to understand the thinking behind the food choices. Interviews with small groups of FEAD recipients may help to examine their perceptions, their degree of satisfaction and better understand their nutritional condition. Finally, the FFQ used although validated in the Greek population comes with limitations in the estimation of fat and salt intake, as discussed elsewhere [12]. Despite potential errors in their accuracy, the use of self-reported anthropometry data and self-reported dietary intakes through an FFQ should not be considered as a substantial source of bias in this analysis, as the main objective was to identify differences between two population groups assessed under the same conditions and using the same tools.

Data on food insecurity in developed countries are extremely rare with the exception of the USA [25,26,27,28], and this study, despite any limitations, is the first of its kind in Greece and the first to assess the effectiveness of the FEAD program in lifting food insecurity in the Greece. In this context it provides crucial details on the needs of a vulnerable population which could be used by policymakers, NGOs and charities in the planning and design of similar interventions.

The evidence provided by the present study could be useful for proposing potential future improvements for the program. The biggest issues are focused on the overall diet quality and energy and protein malnutrition. Firstly, a focus on the distribution of a higher amount of fruits, vegetables and fish/seafood is needed. This will result to better adherence to the guidelines and also address the low fiber intake among FEAD recipients. The low intake of wholegrains among FEAD recipients could also suggest a need for the procurement of wholegrain cereals that are commonly consumed in Greece, such as rusks, wholegrain melba toast and bread, bulgur, etc. As far as the energy malnutrition is concerned, addressing this issue is much more complicated. Given the weight status of the FEAD recipients, one could argue that an energy deficit around 200 kcal per day may even be beneficial for their long-term health; however, our data highlight energy intakes of approximately 500 kcal below the 1950 kcal per day cut off on average. These energy deficits are extreme and should be addressed with the distribution of larger volumes of food provisions per beneficiary. In this process, energy dense and nutrient poor foods should be avoided, and foods included in the food provision should deliver energy, as well as a variety of vitamins and minerals. As FEAD allows for non-material support, the option of nutrition education and cooking classes should be considered in order to educate FEAD recipients about healthy eating habits and provide support in the management of the psychological and social barriers experienced by those living with the stigma of overweight and deprivation. Such changes in the FEAD food provisions are expected to have a direct impact on the dietary habits of those enrolled in the program, especially the provision of food items such as fish, with high retail prices possibly even impacting engagement with the FEAD program. Anecdotal evidence from discussion with the FEAD authorities and recipients indicate patterns of higher engagement with FEAD when items like meat and chicken are distributed, which are considered more expensive compared to food deliveries that include mainly fruits and vegetables. Of course, any changes in the implementation of the FEAD program would require a detailed analysis of the operational guide and other operational data (frequency of food deliveries, procurement data, etc.). Analyzing the perception of FEAD from its recipients and the collection of qualitative data on their experience and potential ways to improve it would also be needed. Such an analysis is currently underway and preliminary data suggest that a stronger focus on fish and seafood and traditional wholegrain-rich foods is possible and would be welcomed by FEAD recipients [46].

## 5. Conclusions

Today, about ten years after the outburst of the economic crisis in Greece, disparities in food accessibility continue to exist. The double burden of malnutrition is becoming evident in vulnerable populations, even after being enrolled in a food assistance program. The diets of those most deprived, despite a positive shift towards more legumes and animal protein sources, continue to include high amounts of meat, confectionary and ice cream, alongside very limited intakes of fruits and vegetables—and an even less fish—altogether providing evidence for a population in a nutritional transition.

## Figures and Tables

**Figure 1 nutrients-11-02914-f001:**
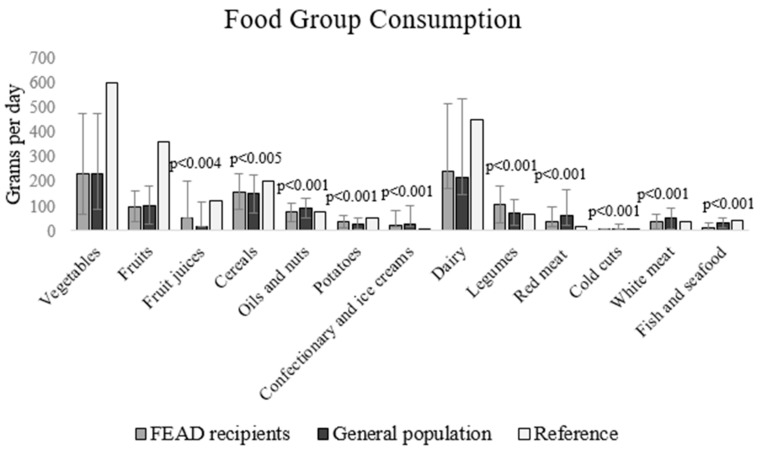
Consumption of food groups in g per day (mean, SD) for FEAD recipients (food insecure) and the general population.

**Figure 2 nutrients-11-02914-f002:**
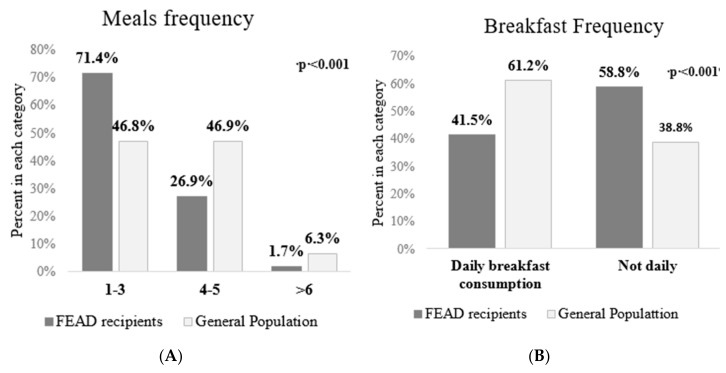
Meals per day (**A**) and breakfast consumption (**B**) comparison between FEAD recipients (food insecure) and general population.

**Figure 3 nutrients-11-02914-f003:**
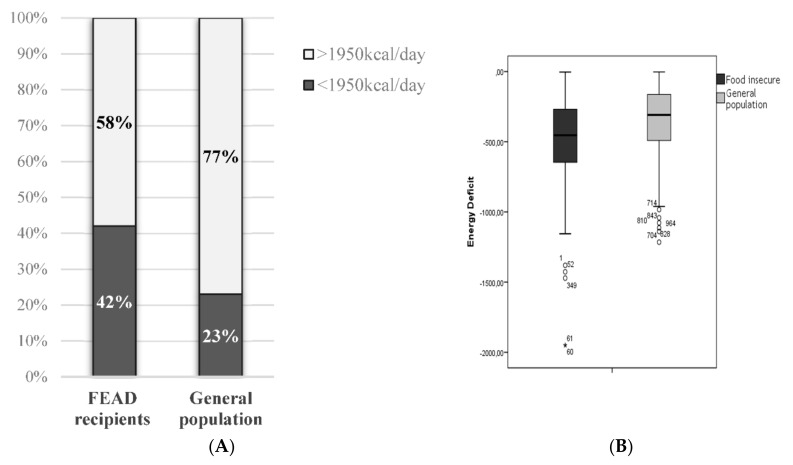
Energy intake percentages between FEAD recipients (food insecure) and the general population, using as the cut-off, 1950 kcal per day (**A**). The box plot illustrates the range of individuals consuming less than the cut off (**B**).

**Figure 4 nutrients-11-02914-f004:**
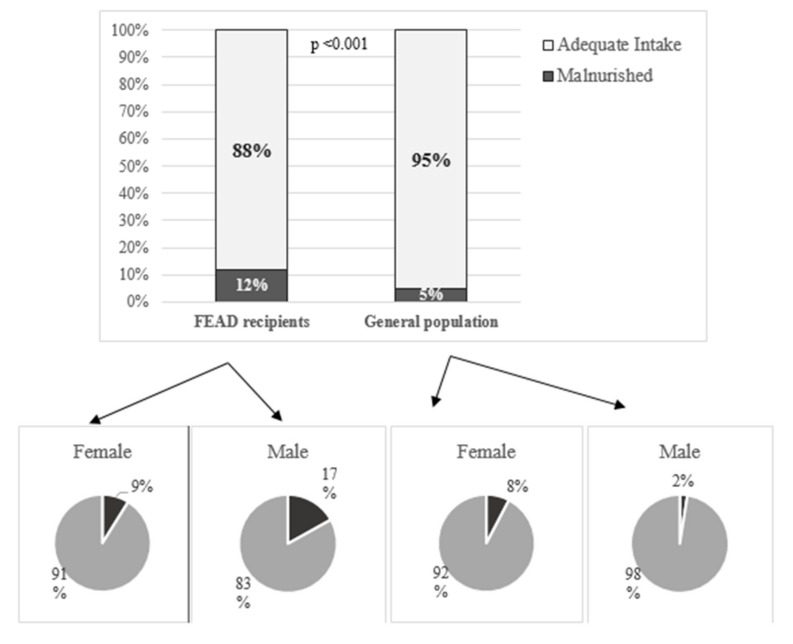
Protein malnutrition percentages (defined as ≤ 0.75 g/kg body-weight) between FEAD recipients (food insecure) and the general population and between genders.

**Table 1 nutrients-11-02914-t001:** Demographic and anthropometric data of participants, which includes food insecure (Fund for the European Aid to the Most Deprived (FEAD) recipients) and the general population. ^a^

	Food Insecure *N =* 499	General Population *N =* 545	*p* ^b^
Female, *n* (%)	301 (60.7)	264 (48.7)	<0.001
Years of age (mean ± SD)	47.53 ± 13.1	47.82 ± 13.6	0.727
Years of education (mean ± SD)	10.98 ± 8.5	12.66 ± 3.6	<0.001
Number of children, *n* (%)			<0.001
0	68 (15.3)	161 (30.9)	
1	107 (24.0)	83 (15.9)	
2	150 (33.7)	209 (40.1)	
3	76 (17.1)	54 (10.4)	
Occupational status, *n* (%)			<0.001
Employed	62 (12.6)	321 (59.3)	
Unemployed	375 (76.4)	84 (15.5)	
Retired	25 (5.1)	94 (17.4)	
Housewife	28 (5.7)	42 (7.8)	
Marital status, *n* (%)			<0.001
Single	123 (24.8)	137 (25.6)	
Married	245 (49.5)	328 (61.2)	
Divorced	105 (21.2)	42 (7.8)	
Widowed	22 (4.4)	29 (5.4)	
Body mass index (kg/m^2^) (mean ± SD)	27.25 ± 5.3	25.93 ± 5.1	<0.001
BMI categories (%)			<0.001
Underweight (<18 kg/m^2)^	1.7%	2.4%	
Normal (18–24.9 kg/m^2^)	28.8%	42.1%	
Overweight (24.9–29.9 kg/m^2^)	44.0%	37.5%	
Obese (> 29.9 kg/m^2^)	25.4%	18.0%	

^a^ Results are presented as means ± SDs, as indicated for normal variables. ^b^
*p*-values derived through the independent sample *t*-test for the normally distributed variables and through the Man–Whitney U-tests for the skewed ones.

**Table 2 nutrients-11-02914-t002:** Intake of macronutrients and indicators of adequate intake between the FEAD recipients (food insecure) and the general population. ^a^

	FEAD Recipients	General Population	*p* ^b^
**Energy (kcal/day)**	2225.3 ± 832.5	2498.2 ± 728.0	<0.001
**Carbohydrates (%E)**	30.92 ± 6.2	27.12 ± 6.0	<0.001
**Protein (%E)**	21.32 ± 7.5	18.07 (15.34, 23.2)	<0.001
**Protein per kg (g)**	1.37 (0.91, 2.09)	1.52 (1.09, 2.10)	0.004
**Protein from plant sources (%E)**	5.81 ± 1.7	4.94 ± 1.3	<0.001
**Protein from animal sources (%E)**	15.59 ± 7.5	15.22 ± 7.1	0.409
**Total fat (%E)**	55.60 ± 8.1	57.52 ± 7.8	<0.001
**PUFA (%E)**	8.00 ± 2.3	7.59 ± 1.9	0.001
**MUFA (%E)**	29.62 ± 6.6	31.48 ± 6.0	<0.001
**SFA (%E)**	14.36(12.18, 18.70)	14.85 (12.85, 20.38)	0.126
**Fiber (g/day)**	27.37(21.77, 45.21)	25.27 (20.04, 31.65)	<0.001
**Calcium (mg/day)**	33.10 ± 15.8	28.03 ± 12.5	0.031
**Sodium (mg/day)**	1037.9 ± 509.3	1245.29 ± 531.81	<0.001

^a^ Results are presented as means ± SDs, as indicated for normally distributed variables, and medians (Q_1_, Q_3_), as indicated for skewed variables. Categorical values were presented as frequency (*n* and relative percent). ^b^
*p*-values were derived through the independent *t*-test for the normally distributed variables and through the Man–Whitney U-test for the skewed ones.

**Table 3 nutrients-11-02914-t003:** Daily intakes of food groups compared with the Greek National Nutritional Guide between FEAD recipients (food insecure) and the general population.^a^

Food Groups	g/day		Portions/day ^c^		WHO Portions	
	FEAD Recipients	General Population	FEAD Recipients	General Population	FEAD Recipients	General Population
**Vegetables**	231.5 (163.4, 244.4)	231.48 (144.0, 244.4)	1.54 (1.09, 1.63)	1.54 (0.96, 1.63)	2.57 (1.82, 2.72)	2.57 (1.60, 2.72)
**Fruits**	98.29 ± 63.7	103.9 ± 78.6	1.03 (0.66, 1.03)	1.03 (0.22, 1.03)	1.38 (0.88, 1.38)	1.37(0.29, 1.38)
**Fruit juices**	50.4 (0.0, 153.6	16.8 (0.00, 102.0) *	0.40 (0.00, 1.23	0.13 (0.00, 0.82) *	0.40 (0.00, 1.23	0.13 (0.00, 0.82) *
**Cereals**	158.58 ± 70.85	149.7 ± 77.2			5.29 ± 2.4	4.99 ± 2.6 *
Pasta, rice			1.88 (1.33, 2.31	1.33 (0.90, 2.31) *		
Bread			1.00 (1.00, 2.00)	1.00 (0.64, 2.00)		
**Dairy**	244.0 (70.44, 274.0)	214.98 (70.44, 319.9)	1.22 (0.35, 1.37)	1.07 (0.35, 1.60)		
Cheese					0.38(0.13, 0.60	0.60 (0.60, 0.60) *
Milk					1.00(0.21, 1.01)	0.64(0.21, 1.21)
**Olive & nuts**	75.8 ± 37.2	91.96 ± 38.4 *				
Oils			4.68 (3.33, 6.82	6.67 (3.49, 6.82) *	4.68 (3.33, 6.82	22.5 (11.76, 23.03) *
Nuts			0.22 (0.00, 0.67	0.67 (0.00, 2.04) *	0.23 (0.00, 0.070	0.70 (0.00, 2.13) *
**Legumes**	104.96 ± 74.5	73.6 ± 53 *	0.42 (0.42, 1.28	0.42 (0.42, 0.42) *	2.10 (2.10, 6.40	2.10 (2.10, 2.10) *
**Red meat**	38.85 (19.95, 59.85	59.85 (38.85, 105.98) *	0.32 (0.17, 0.50	0.50 (0.32, 0.88) *	1.30 (0.67, 2.00	2.00 (1.30, 3.53) *
**Cold cuts**	2.10 (0.00, 6.30	6.30 (0.00, 19.20) *	0.11 (1.09, 1.63	0.32 (0.00, 0.96) *	a.c. ^e^	a.c. ^e^
**White meat**	37.7 ± 29.6	51.2 ± 43.1*	0.26 (0.26, 0.26	0.26 (0.26, 0.80) *	1.05 (1.05, 1.05	1.05 (1.05, 3.20) *
**Eggs**	22.09 ± 19.6	20.03 ± 21.5	0.24 (0.24, 0.73	0.24 (0.24, 0.73) *	0.24 (0.24, 0.73	0.24 (0.24, 0.74) *
**Fish & seafood**	10.5 (10.5, 31.5	31.5 (10.5, 42.0) *	0.07 (0.07, 0.21	0.21 (0.07, 0.28) *	0.35 (0.35, 1.05	1.05 (0.35, 1.40) *
**Potatoes**	38.16 ± 25.7	29.1 ± 23.7*	0.43 (0.14, 0.43	0.14 (0.14, 0.43) *	1.31 (0.43, 1.31	0.43 (0.43, 1.31) *

^a^ Results are presented as means ± SDs, as indicated for normally distributed variables, and median (Q_1_, Q_3_), as indicated for skewed variables. ^b,^* *p*-value < 0.05 derived through the independent sample *t*-test for the normally distributed variables and through the Man–Whitney U-test for the skewed ones. ^c^ Portions per day as defined by National Nutritional Guide of Greece. ^d^ Portions per day as defined by WHO Eastern Mediterranean Region Office. ^e^ a.c. = avoid consumption.

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
