# Peer review of "Nutrition Transition in the Post-Economic Crisis of Greece: Assessing the Nutritional Gap of Food-Insecure Individuals. A Cross-Sectional Study"

_nutrients, 2019, doi:10.3390/nu11122914_

Round 1
Reviewer 1 Report
The authors present an interesting study in which evaluated the nutritional intake and dietary habits of FEAD recipients in Greece with the objective to better understand the needs and habits of this vulnerable population group and help improve local policies and public health actions in the framework of the FEAD program in Greece.
The manuscript is properly structured and drafted.
The Introduction section adequately contextualizes the problem, providing specific data on the risk nutritional situation among the Greek population.
The Methods section adequately describes the participant selection process, inclusion criteria, and the final sample size. However, the authors should improve the following aspects:
In the section “Research Design and inclusion exclusion criteria of study sample”, the authors must include a flow chart to summarize the participants' recruitment process.
Page 3, lines 104-107: The authors indicate “… All data were collected via a questionnaire which collected self-reported data on socio-demographic characteristics, dietary habits and anthropometry via a 20-minute interview with a trained nutritionist”. How researchers controlled the accuracy of the information provided by participants? This aspect could condition all study results. Page 3, line 114: The authors indicate “… Body weight (in kilograms) and height (in meters) were recorded as self-reported values”. However, this information is very inaccurate and does not allow to estimate the weight and height of the subjects accurately. Therefore, BMI values may be inaccurate conditioning the results obtained in categories (%) underweight, normal, overweight and obese. What criteria were used to define overweight and obesity? WHO criteria?
The Results section clearly describes the main findings of the study. The tables are pertinent and improve the compression of the contents.
The Discussion section is consistent with the results. The authors discuss adequately the results obtained, comparing with studies of similar characteristics.
In Conclusion section, authors should indicate the importance of these results for the implementation of future programs to improve the nutritional health of the Greek population.
The bibliography consulted is pertinent.
Author Response
Reviewer 1 Comments
The authors present an interesting study in which evaluated the nutritional intake and dietary habits of FEAD recipients in Greece with the objective to better understand the needs and habits of this vulnerable population group and help improve local policies and public health actions in the framework of the FEAD program in Greece. The manuscript is properly structured and drafted. The Introduction section adequately contextualizes the problem, providing specific data on the risk nutritional situation among the Greek population. The Methods section adequately describes the participant selection process, inclusion criteria, and the final sample size. The Results section clearly describes the main findings of the study. The tables are pertinent and improve the compression of the contents. The Discussion section is consistent with the results. The authors discuss adequately the results obtained, comparing with studies of similar characteristics. In Conclusion section, authors should indicate the importance of these results for the implementation of future programs to improve the nutritional health of the Greek population. The bibliography consulted is pertinent.
However, the authors should improve the following aspects:
Comment 1: In the section “Research Design and inclusion exclusion criteria of study sample”, the authors must include a flow chart to summarize the participants' recruitment process.
The whole recruitment took place in two waves, first cases and then controls, in a single step process where all participants were invited to participate to the study, checked for inclusion/exclusion criteria and enrolled in the study. A comment has been added in the methods section. Due to the single step process it was not possible to create a flow chart. (Lines 110-111)
Comment 2: Page 3, lines 104-107: The authors indicate “… All data were collected via a questionnaire which collected self-reported data on socio-demographic characteristics, dietary habits and anthropometry via a 20-minute interview with a trained nutritionist”. How researchers controlled the accuracy of the information provided by participants? This aspect could condition all study results.
The questionnaire used has been previously developed and validated for its accuracy under the same research conditions. Although self-reported values are always prone to misreporting this is likely to be the same among study participants and we do not expect the questionnaire used to be a significant source of bias when performing comparisons between the two groups. A comment has been added in the discussion (Lines 288-295 & 333-335).
Comment 3: Page 3, line 114: The authors indicate “… Body weight (in kilograms) and height (in meters) were recorded as self-reported values”. However, this information is very inaccurate and does not allow to estimate the weight and height of the subjects accurately. Therefore, BMI values may be inaccurate conditioning the results obtained in categories (%) underweight, normal, overweight and obese. What criteria were used to define overweight and obesity? WHO criteria?
The use of self-reported body weight and height values is indeed prone to misclassification of BMI values. It is true that in cross-sectional studies aiming to identify differences between BMI categories such an issue could skew the results, but the nature of our analysis is comparative between two population groups. In this context there is no indication that the misreporting of body weight and height is substantially different in the two population groups (FEAD recipients and general population) and so we believe that the conclusions on the relative difference in BMI categories prevalence are valid and unbiased. A relevant comment has been added in the discussion section (Lines 341-344).
Reviewer 2 Report
Aim: assess nutritional intake and dietary habits for FEAD recipients in Greece with objective to better understand needs and habits of this vulnerable population group and help improve local policies and public health actions in the framework of the FEAD program in Greece
Does the exclusion of subjects who do not speak Greek eliminate the study of immigrant/refugee populations which may also be at risk for food insecurity? Similarly, individuals who do not live in cities?
Why is anthropometric data self-reported? This could severely impact the findings.
It would be helpful to provide details of the recommendations from FAO/WHO that were used.
Table 3 – data organization makes direct comparisons difficult, consider listing the data in an order like FEAD gr/day, General Population gr/day, FEAD portions/day, General Population portions/day, etc.
Table 3 – why is the only data set to be statistically analysed the gr/day data? If that is the most important data set, why include the others? If the others are important, why not analyse them?
Figure 1 – What do the error bars represent? SD? SEM?
In the results, it is reported that FEAD recipients have a higher intake of PUFA and lower intake of MUFA than the general population (pg 5, line 184), but the discussion states that they have a lower intake of PUFA (pg 10, line 253).
Why is the data regarding weekly fish intake not shown or included in supplementary data?
The omission of table salt from the questionnaire is a significant oversight. It’s good to see it acknowledged, but is there any way to re-assess for this data?
The impact of this study would be greatly increased if the authors were able to draw conclusions about what the differences in health outcomes related to diet are for each population. For example, do the more obese, food insecure individuals have increased rates of hypertension, diabetes etc.?
There does not appear to be a clear conclusion with respect to the objectives set forth in the introduction of this paper. If the aim was to gain understanding with the outlook to improve policy, this should be re-addressed with ideas about how this information is helpful and can be used. What are changes that the authors might recommend for the FEAD program? What are the main contributing factors in the diets of individuals on FEAD that can be changed to improve weight status?
Author Response
Reviewer 2 Comments
Aim: assess nutritional intake and dietary habits for FEAD recipients in Greece with objective to better understand needs and habits of this vulnerable population group and help improve local policies and public health actions in the framework of the FEAD program in Greece
Comment 1: Does the exclusion of subjects who do not speak Greek eliminate the study of immigrant/refugee populations which may also be at risk for food insecurity? Similarly, individuals who do not live in cities?
The questionnaire was indeed designed in Greek and not translated to other languages. The study population however was not exclusively Greek citizens. The local authorities and the FEAD managing authority had requested that the questionnaire do not include extensive information on socioeconomic status as well as country of origin to avoid stigma and the creation of an environment of social inequalities. According to previous reports however FEAD Greece had reported that only 1% of the beneficiaries in 2016 were migrants. This could be due to the nature of the FEAD food provision program which is available only to individuals who have been properly integrated in the Greek social system (national insurance number, Visa etc). Despite the high risk of food insecurity in this group a different study will be needed to quantify this risk, one aimed to recruit participants in the entrance points for refugees. Finally, in regards to the recruitment of individuals who do not live in the cities, the study sample was recruited in five administrative areas across Greece. It included urban and rural population, with recruitment taking place in Athens and Thessaloniki (urban environment) as well as smaller cities and towns in Greece. Relevant comments have been added in the methods sections as well as in the discussion (Lines 98-99, 106-107, 323-329).
Comment 2: Why is anthropometric data self-reported? This could severely impact the findings.
The use of self-reported body weight and height values is indeed prone to misclassification of BMI values. It is true that in cross-sectional studies aiming to identify differences between BMI categories such an issue could skew the results, but the nature of our analysis is comparative between two population groups. In this context there is no indication that the misreporting of body weight and height is substantially different in the two population groups (FEAD recipients and general population) and so we believe that the conclusions on the relative difference in BMI categories prevalence are valid and unbiased. A relevant comment has been added in the discussion section (Lines 341-344).
Unfortunately, the measurement of weight and height at the time of the interview was not possible. As mentioned in the Methods, recruitment and data collection took place at the time of the food provision delivery. Food provisions usually take place in warehouse with no access to dedicated areas for the measurement of weight and height in a way that respects the sensitive nature of these data. Additionally, although actual measurement of weight and height would improve accuracy it would add a burden to the participants (whose main concern is the timely and efficient collection of the food provision) and the staff working in the food provision delivery site.
Comment 3: It would be helpful to provide details of the recommendations from FAO/WHO that were used.
A brief description of the food based guidelines used was added in the Methods section. (Lines 160-165)
Comment 4: Table 3 – data organization makes direct comparisons difficult, consider listing the data in an order like FEAD gr/day, General Population gr/day, FEAD portions/day, General Population portions/day, etc.
The table has been formatted in order to address this comment (Table 3, revised)
Comment 5: Table 3 – why is the only data set to be statistically analysed the gr/day data? If that is the most important data set, why include the others? If the others are important, why not analyse them?
All data has been analyzed statistically. P-values for all datasets are added to Table 3. The use of gr/day as the main outcome was chosen as it represents more clearly the differences in food choices between the two groups. The analysis as servings/day is conducted to test adherence to the guidelines and hence diet quality. The statistical analysis indicates very similar results among the three approaches.
Comment 6: Figure 1 – What do the error bars represent? SD? SEM?
All error bars represent SD. A comment has been added in the figure
Comment 7: In the results, it is reported that FEAD recipients have a higher intake of PUFA and lower intake of MUFA than the general population (pg 5, line 184), but the discussion states that they have a lower intake of PUFA (pg 10, line 253).
We apologise for the error and have corrected it. (Lines 266-271)
Comment 8: Why is the data regarding weekly fish intake not shown or included in supplementary data?
The supplementary table has been formatted accordingly. The WHO Eastern European Region discusses meat, fish and beans include under the scope of iron intake and proposes a general target of 160gr per day for all of those foods. Later it proposes a frequency for red meat and white meat. Fish is discussed separately and a recommendation for two 180g portions per week is made. The table has been formatted to incorporate this information as opposed to the previous version that only presented the recommendation for combined intake.
Comment 9: The omission of table salt from the questionnaire is a significant oversight. It’s good to see it acknowledged, but is there any way to re-assess for this data?
Unfortunately, this is an inbuilt error of the questionnaire used and cannot be addressed at this point. However, as the same questionnaire has been used almost all studies carried in Greece the past decade the results presented in this manuscript are directly comparable to the literature. This includes previous study conducted by our team with the aim to evaluate salt intake in Greece, already mentioned in the manuscript as reference 30. The issue has been included in the discussion alongside a comment for the need of a single common FFQ validated in the Greek population for the measurement of salt intake (Lines 288-295).
Comment 10: The impact of this study would be greatly increased if the authors were able to draw conclusions about what the differences in health outcomes related to diet are for each population. For example, do the more obese, food insecure individuals have increased rates of hypertension, diabetes etc.?
We agree with the reviewer on the added value that linking eating habits to health outcomes, however no such data were collected as part of this study. The main reason behind this being an attempt to not overwhelm study participants and avoid the collection of a large list of personal data which could hinder the recruitment and the willingness of the FEAD partners to provide access to their sites and beneficiaries. As the first study of its nature in Greece we believe that the publication of these results could support the need for further research and the collection of additional data.
Comment 11: There does not appear to be a clear conclusion with respect to the objectives set forth in the introduction of this paper. If the aim was to gain understanding with the outlook to improve policy, this should be re-addressed with ideas about how this information is helpful and can be used. What are changes that the authors might recommend for the FEAD program? What are the main contributing factors in the diets of individuals on FEAD that can be changed to improve weight status?
The discussion has been expanded to include a section with clearer proposals on the improvement of the FEAD program (Lines 345-363).
Round 2
Reviewer 1 Report
No comment
Author Response
We would like to thank the reviewer for the time and effort spent throughout this process.
Reviewer 2 Report
The corrections listed address initial concerns about the thoroughness of the study, however, the paper does have many limitations that cannot be overcome in terms of the way that data was collected. In its current state, this manuscript is acceptable, however the novelty of this study and the importance of these findings are not adequately emphasized. While the public health and policy implications are addressed in this draft where they weren't before, the conclusions offered aren't specific to the findings and outcomes. How might the changes proposed influence specific markers? Are the proposed changes realistic for the FEAD program?
Author Response
On behalf of the author team I would like to thank you for considering our paper titled: “Nutrition transition in the post-economic crisis Greece: Assessing the nutritional gap of food insecure individuals. A case control study.” for publication.
We are grateful for the reviewers’ valuable time and comments and we believe the following letter will successfully address the points raised.
The corrections listed address initial concerns about the thoroughness of the study, however, the paper does have many limitations that cannot be overcome in terms of the way that data was collected. In its current state, this manuscript is acceptable, however the novelty of this study and the importance of these findings are not adequately emphasized. While the public health and policy implications are addressed in this draft where they weren't before, the conclusions offered aren't specific to the findings and outcomes. How might the changes proposed influence specific markers? Are the proposed changes realistic for the FEAD program?
We would like to thank the reviewer for the comment. We have updated the relevant section of the discussion to include further information why we believe this study is novel and important in its filed. We have also provided more tailored suggestions on how FEAD could change with an aim to see improvements in dietary habits. Unfortunately, we believe that the study of any other markers of disease would be too complicated as they can be affected by a multitude of factors (smoking habits, homeliness, personal hygiene, oral & sexual health, drug use etc). At its current state FEAD does not collect any data on those other factors and it is not intended to in order to limit social inequalities. At this stage our group is conducting a detailed analysis of the FEAD operational guide coupled with procurement data to assess the implementation of FEAD and suggest specific changes. We are also analyzing data from FEAD recipients in order to offer an end-user evaluation of the program which can be further used as an inspiration for changes. Lines 347-351 & Lines 369-380